# Alternative Biochemistries for Alien Life: Basic Concepts and Requirements for the Design of a Robust Biocontainment System in Genetic Isolation

**DOI:** 10.3390/genes10010017

**Published:** 2018-12-28

**Authors:** Christian Diwo, Nediljko Budisa

**Affiliations:** 1Institut für Chemie, Technische Universität Berlin Müller-Breslau-Straße 10, 10623 Berlin, Germany; christian.diwo@tu-berlin.de; 2Department of Chemistry, University of Manitoba, 144 Dysart Rd, 360 Parker Building, Winnipeg, MB R3T 2N2, Canada

**Keywords:** alternative amino acid and nucleotide repertoires, alternative core cellular chemistries, biocontainment, genetic firewall, genetic isolation, orthogonal central dogma of molecular biology, synthetic life, xenobiology

## Abstract

The universal genetic code, which is the foundation of cellular organization for almost all organisms, has fostered the exchange of genetic information from very different paths of evolution. The result of this communication network of potentially beneficial traits can be observed as modern biodiversity. Today, the genetic modification techniques of synthetic biology allow for the design of specialized organisms and their employment as tools, creating an artificial biodiversity based on the same universal genetic code. As there is no natural barrier towards the proliferation of genetic information which confers an advantage for a certain species, the naturally evolved genetic pool could be irreversibly altered if modified genetic information is exchanged. We argue that an alien genetic code which is incompatible with nature is likely to assure the inhibition of all mechanisms of genetic information transfer in an open environment. The two conceivable routes to synthetic life are either de novo cellular design or the successive alienation of a complex biological organism through laboratory evolution. Here, we present the strategies that have been utilized to fundamentally alter the genetic code in its decoding rules or its molecular representation and anticipate future avenues in the pursuit of robust biocontainment.

## 1. Introduction

The design and manufacturing of specialized biological systems promises to be the next giant leap in human technology to advance many aspects of our society in unimaginable ways. Technologies derived from this scientific groundwork are expected to be employed for fine chemical and pharmaceutical production [1,2], transform medicine and epidemic control [3,4], be employed for environmental remediation [5], mining [6] and crop fertilization [7], access novel renewable energy sources [8,9,10], and complement electronic circuits and computational devices [11,12].

The modification of metabolic pathways as well as the addition, deletion, minimization or integration of gene circuits (genes, gene clusters) transforms living organisms into autonomously acting tools [13] able to execute preprogrammed processes, generally termed genetically modified organisms (GMOs) [14]. The modern laboratory methods of synthetic biology (SB) take advantage of the modularity of living systems, mixing and matching traits from various species in order to create organisms with specific desired functionality [15]. The methods of SB aiming to transfer naturally evolved functions between organisms can be distinguished from the emerging field of xenobiology (XB), which aims to expand the framework of natural chemistries within living cells through the incorporation of non-natural building blocks [16,17].

Indeed, life on Earth is a reservoir of complex organized systems, displaying different levels of control over their respective dynamic environments and intricate physiological processes to achieve robust autonomy [18,19]. In this context, the genetic information encoding the organization of an organism can persist in two ways, either within a species through reproduction (vertical gene transfer, VGT), or between species through horizontal gene transfer (HGT). The pool of all possible manifestations of a certain mechanism of control that can be acquired by mutation and selection drastically expands with the complexification of biological systems. Thus, the exchange of components of the cellular organization from one path of evolution to another potentially can advance certain aspects of control which leads to a more robust population [20,21]. Exchanging information between species at the genetic level is only possible for individuals maintaining the same physicochemical character (DNA and RNA) and decoding logic (transcription and translation) in their genetic code, making the genetic code the lingua franca of life on Earth [17].

However, it is the universality of the genetic code which makes GMOs designed to be employed in open environments a potential threat, as leakage of synthetic genetic information polymers might contaminate the naturally evolved genetic pool and alter entire ecosystems [4]. There is an overwhelming consensus in both the public and scientific community that potentially harmful consequences of GMOs must be prevented by engineering appropriate safety measures before they can be safely employed [22,23]. Thus, the aim for a robust biocontainment system should be to achieve control over all possible mechanisms of proliferation of genetic information, which comprises preventing unintended VGT and HGT, as well as the possibility of circumvention of the biocontainment due to loss of genetically encoded safety mechanisms (genetic drift) [24]. Linking the expression of essential genes to the external supply of synthetic molecules has achieved a relatively high level of control over the reproduction of single-celled organisms [25]. The level of safety is usually measured in cells escaping the containment relative to the total cell count in a population. A biocontainment system is regarded as safe below 1 escapee in a population of 10^8^ cells, which is the safety threshold proposed by the National Institutes of Health (NIH) [26]. Introducing dependency on synthetic molecules into the molecular complexes and processes along the flow of genetic information (central dogma) [27] seems to be an attractive target, directly impacting the universal characteristics of the genetic code. For example, the development of alternative genetic information storage molecules (xeno-nucleic acids = XNAs), or the alteration of the universal decoding logic through systematic introduction of non-canonical amino acids, should enable the development of biocontained synthetic organisms (Figure 1). 

Tight containment of artificially altered genetic information is the primary concern of a biocontainment system. For such a system to be employed by industry, it must also enable standard engineering methodologies that do not impair the development of viable products [23]. Industrial needs are a legitimate aspect of a biocontainment system to be considered. However, this essay will be mainly concerned with the basic requirements and paths of implementation explored to create life in genetic isolation. Microorganisms are the preferred scaffold for developing biological tools (e.g., *E. coli*, *B. subtilis*, *S. cerevisiae*), where natural processes can be redesigned over relatively short timescales through the methods of SB and XB. As such, biocontainment concepts applied to single cellular organisms are presented. Approaches not concerned with altering the core chemistries of cellular organization are summarized elsewhere [28,29].

## 2. Basic Considerations for an Alien Central Dogma

The cellular organization of even the simplest organisms is immensely complex and interconnected, to which the central dogma is an interpretation of the flow of information, illustrating a general theme conserved in every modern organism [27]. The molecular outputs (RNAs, proteins) along the ribosomal protein biosynthesis and their transforming processes (transcription, translation) possess a multitude of chemical and physical identities which are heavily interconnected [34], regulating the cellular processes over the complete cell cycle and in response to environmental cues. The topology of this network (who interacts with whom) is crucial for a cell to maintain robust autonomy [35] and manipulating these intricate associations for the purpose of biocontainment is a demanding task but thought to be very effective [28,36]. VGT is based on cellular reproduction, and as such depends on many synchronized processes in order to guarantee transmission of the complete genetic information to a robust next generation. Thus, intercepting any process along the central dogma will result in the cell losing its autonomy and prevents cell replication. For example, introducing dependencies to synthetic compounds into components of the central dogma results in trophic containment controlling cell survival through the external supplementation of the compound [24]. HGT, in turn, is a type of communication between cells and relies on the universality of the genetic code in its decoding logic and its physical representation. The main vectors of HGT between prokaryotes are transduction, conjugation or DNA uptake from the environment [37]. For successful HGT to take place, a cell is required to replicate or recombine, transcribe and translate received genetic information molecules into functional proteins and thus, HGT can be ruled out by altering the decoding rules (meaning), or the genetic information storage molecules (identity) of the genetic code [38]. 

## 3. Altering the Meaning of the Genetic Code

The idea of genetically encoding alternative cellular chemistries by incorporating non-canonical amino acids (ncAAs) into the proteome of microorganisms has spawned efforts to free codons from their canonical assignment and engineer the cellular translation machinery to accept synthetic compounds [39]. Genes harboring an alternative codon assignment will not be expressed in a cell with a standard codon assignment receiving the altered information, and thus the genetic information will be contained (Figure 2).

DNA is encoding the cellular organization through several layers of its chemical identity. Codons are the logic units receiving their meaning through the coupling of a specific amino acid to a cognate tRNA adapter with the correct anticodon, a reaction catalyzed by a tRNA- and amino acid-specific aminoacyl-tRNA synthetase (aaRS). Eighteen out of 20 amino acids are encoded by multiple synonymous codons. Rather than being a redundancy, the degeneracy of the genetic code—61 codons encoding 20 + 2 amino acids and 3 codons encoding stop signals—allows for decision-making processes rooted at a basic level, where the codon distribution is directly informing translation processes at the ribosome, or gene regulation processes [40]. Direct reassignment of a codon will change all instances of the previously assigned amino acid in the proteome. The misincorporation of an amino acid at the reassigned codon will typically lead to misfolded and non-functional proteins [41], while the genome-wide substitution of a codon by a synonymously coding triplet is likely to drastically influence the cellular organization [40,42]. In recent decades, several schemes have been devised to overcome the challenges of cellular complexity and achieve partial genetic isolation through alteration of the canonical codon assignment.

### 3.1. Stop Codon Reassignment

Stop codon suppression (SCS) exploits the low abundance of the amber stop codon (UAG) in the *E. coli* genome for its reassignment into a sense codon. The introduction of an orthogonal tRNA:aaRS pair into the cell is used to facilitate the incorporation an ncAA at the command of the reassigned codon. However, with the functional canonical process in competition with the engineered assignment, a code ambiguity is created. The genome-wide substitution of the UAG codon with a synonymous codon allows for the deletion of the competing release factor (RF1), which mitigates the detrimental effects of the code ambiguity and enhances the ncAA incorporation efficiency [43]. Using this strain as a platform, highly contained organisms have been created through the design of essential proteins with functional dependencies on ncAA incorporation [24,41]. The design and selection of the proteins with the desired properties demands sophisticated methodologies, as incorporating UAG codons into genes might also influence mRNA secondary structures and ribosome-binding strength [44]. Once robust dependencies are found, combining several mutated sites into a few essential genes in one organism results in exceptionally low escape frequencies well below the suggested NIH threshold [41]. Essential for the robustness of the system are both the selectivity of the aaRS [45] and the tolerance of the protein towards regaining its function through misincorporation of canonical amino acids. Interestingly, Tack and colleagues developed a portable biocontainment system demonstrated on strains of *E. coli* and several other species of bacteria sensitive to ampicillin by modulating an ampicillin resistance gene to only be functional upon ncAA incorporation [46]. By supplying the antibiotic resistance gene and the tRNA:aaRS pair on a plasmid, they managed to contain the tested organisms over several hundred generations with escape rates one order of magnitude lower than the NIH threshold. This data suggests that genetic code ambiguity does not effectively reduce the containment if linked to a strong impact on cell survival. 

While persistent in vivo incorporation of ncAAs into proteins has been achieved over many generations, the level of containment has yet to be improved [47]. The reassignment of a single non-sense codon can be readily overcome by evolution [24,41], or post-transcriptional modification (although demonstrated for a reassigned sense codon) [48] of a tRNA incorporating canonical amino acids at the command of the UAG codon. Such genes could in principle also be acquired by HGT. Although initially the UAG and RF1 deficient strain is less suitable as phage hosts [49], bacteriophage T7 has been able to adapt, overcoming the immunity of cells with an expanded genetic code [50]. This mechanism of immunity is traced back to a rescue pathway for stalled ribosomes and deleting the corresponding gene obliviates the immunity conferred by the UAG stop codon deletion [51]. The dependency of essential proteins to an additional ncAA through the reassignment of a second stop codon is conceivable, but potentially highly toxic for the cell [52].

### 3.2. Sense Codon Reassignment

A naturally evolved sense codon reassignment, exchanging the codons assigned to amino acids with entirely different chemical characters (Leu(CUG)Ser, non-polar to polar), does occur in a yeast fungus. Heterologous expression of genes containing the CUG codon in a similar organism with a standard genetic code leads to misfolded proteins, and thus the information transfer cannot be completed [53]. These findings suggest that the reassignment of a sense codon can be used to alter the cellular biochemistry on a proteome scale, ultimately interrupting HGT. The choice of the target codon, either degenerate or exclusively encoding an amino acid, will respectively expand the amino acid repertoire or substitute a canonical building block. However, there are currently no real examples of biocontainment achieved through a sense codon reassignment to an ncAA. The narrow choice of analogs which are tolerated for a proteome-wide incorporation limits the dependency of an organism to an ncAA.

It is assumed that robust reassignment of a sense codon can, in principle, be achieved by a small number of successive topology changes [35]. However, the exact mechanism of how variant genetic codes arise remains elusive and is subject to ongoing investigations [54]. Due to the lack of deep insights into mechanisms of cellular network plasticity [55], changes in the protein biochemistry on a proteome scale are usually invoked through laboratory evolution, forcing the cell to incorporate an increasing amount of an ncAA in order to survive [56]. Several long-term evolution experiments have yielded strains which are adapted to the complete replacement of tryptophan (encoded by a single codon) with a close chemical analog, whereas their ancestors are not able to show any growth under the same conditions [32,57]. However, more interesting for the purpose of biocontainment is to drive the cellular organizational topology beyond the ability to accept the canonical amino acid. A strain of *B. subtilis* has evolved to grow exclusively on several fluorinated tryptophan analogs, whereas it has lost the ability to accept the canonical amino acid tryptophan [58]. The adaptation is traced back to a mutated transporter and simply reverting a single mutation leads to a reconstitution of the ability to use the canonical amino acid [59]. Nevertheless, these findings indicate that obliviating the dependency of an organism for a canonical amino acid enables evolutionary processes to alter the cellular organization, which might lead to incompatibility for the canonical substrate.

In order to capture the assignment of a degenerate codon, highly codon-specific translation capabilities have to be added to an organism. Wobble pairing of canonical base pairs, as well as post-transcriptional modifications can alter the specificity of the codon–anticodon matching [60,61]. Several investigations have tackled the rare, degenerate, codon AGG to code for an ncAA by introducing an orthogonal tRNA:aaRS pair [62,63]. Bröcker and colleagues demonstrated that the specific requirements for selenocysteine translation (mRNA hair pin structure and special elongation factor [64]) can completely overwhelm the natural codon assignment when expressed in competition [65]. In order to completely eliminate ambiguous decoding of a captured codon, Bohlke and Budisa suggest exploiting the decoding mechanism of the AUA codon via a post-transcriptionally modified tRNA [61]. However, successful reassignment of the AUA codon has yet to be reported, whereas the AGG codon has been successfully reassigned to an ncAA. 

These works demonstrate that, in principle, sense codon reassignments with close analogs are possible. However, the currently missing insight into which of the topology changes that have surfaced are necessary to accommodate alternative amino acids result in a lack of reproducibility and portability. Given enough time and data, these attempts may possibly lead to the discovery of fundamental engineering principles enabling the permanent alienation of the cellular biochemistry on a proteome scale [66].

### 3.3. Minimal Cell Design

Cellular organization has evolved employing the 20 canonical amino acids and their codons to confer maximal robustness [67]. However, under laboratory conditions many of the preprogrammed cellular responses might not be necessary, or be rather hindering for certain engineering approaches [68]. Thus, reducing the cellular complexity, by either reducing the size of the genome or by decreasing the degeneracy of the genetic code might help to uncover design principles and to simplify engineering approaches. Additionally, reducing the genetic code degeneracy may free multiple codons for the reassignment to ncAAs [47]. There has been noteworthy progress towards the construction of a so-called minimal cell (genome reduction and sense codon liberation), demonstrating the feasibility of computationally redesigning entire genomes.

In their latest iteration of a minimal genome, the Venter research group created a viable cell with a genome size of 531,000 base pairs named syn3.0, reducing the genome size of *M. mycoides* to approximately half [33]. Of the 473 genes encoded in the genome of syn3.0, 149 are of unknown function, but all seem essential in conditions considered to be optimal for cell growth (no competition for food, optimal temperature, supply of all small molecules in the medium).

Aiming to decrease the genetic code degeneracy, Lajolie and colleagues have devised an algorithm which replaces all instances of 13 rare codons in the 41 ribosomal protein-coding genes [40]. In 2016 they broke further ground by demonstrating the deletion of 7 rare codons, independently on each of 55 stretches of the *E. coli* genome (each ca. 1% of the genome) within certain constrains (for example, conserving relative codon usage, ribosomal binding sites and mRNA secondary structure) [42]. Due to the careful selection of codons, which might in any case be subject to low fidelity decoding due to wobble pairing, only limited growth defects and variations in transcription levels of affected genes can be observed [69].

Eliminating the ability of a cell to decode certain codons improves the genetic isolation, as foreign DNA from viruses, plasmids or other vectors can no longer be properly expressed, thus limiting the potential of HGT [42]. On the other hand, cells equipped with a minimal genome might be useful as a platform host to accept additional contained genetic information. An orthogonal central dogma (orthogonal replication, transcription and translation) could serve as a modular system to equip cells with some desired functionality [36]. A bottleneck to further investigation is the ability to incorporate large segments of synthetic DNA into the genome of living organisms or the creation of entire synthetic genomes. However, a more complete summary of the advances in the field of synthesizing and implementing whole genomes has been reviewed elsewhere [70,71]. 

## 4. Altering the Identity of the Genetic Code

Rather than trying to expand the cellular biochemistry through the replacement of preexisting coding events, researchers are exploring the feasibility of expanding the coding capacity of the genetic code. Codons are the naturally evolved coding units, with a coding capacity limited to 64 variations restricted by the four different nucleotides binned into units of three. Recent investigations have tested the ability to expand either the nucleotide base repertoire, or the codon unit to a higher binning. Any alteration of the fundamental coding units is likely to be poorly tolerated by the natural translation apparatus [72] or likely to fail the canonical replication and transcription machinery, thus leading towards the creation of a genetic firewall (Figure 3).

### 4.1. Frameshift Codons

The advantages of a quadruplet code towards the creation of a genetic firewall seem obvious, in conferring a dramatic change to the fundamental coding identity of the genetic code, through a simple change in the unit convention brought about by only a few altered cellular components. Here, frameshift suppression is classified as a change in genetic code identity due to the expansion in coding capacity, although in order to convey these changes, the physical representation of the genetic information storing polymers does not need to be altered. In 2010 the group of Chin engineered a prokaryotic ribosome able to decode quadruplet codons, introducing an orthogonal in vivo translation system, which in theory would allow for 256 independent coding variants [73]. Linking the survival of a cell to the correct decoding of the AAGA codon, Neumann and colleagues discovered an evolved ribosome, with the ability to polymerize an ncAA in response to both the quadruplet codon and the UAG codon. Relying entirely on the promiscuity of canonical ribosomal protein biosynthesis, the group of Schulz found an orthogonal tRNA:aaRS pair able to efficiently decode a quadruplet codon without severe growth defects for the cell [74]. Expanding the codon capacity to four bases does not appear to be the limit. Hohsaka and colleagues demonstrated that a five-nucleotide based coding units can be considered plausible [75].

While in principle a quadruplet-based genetic code is possible, shortcomings in the understanding of genome engineering as explained in the Section 3.3 “Minimal Cell Design” hinder the progress towards compiling whole genomes. In order to enable the evolvability of ribosomes, they initially have to be freed from their natural duty by finding an orthogonal mRNA:ribosome pair [72]. The reversion of such a containment through HGT or evolution seems unlikely due to the highly toxic effects of a translation process that would be able to decode both canonical triplet and quadruplet codons simultaneously.

### 4.2. Synthetic Base Pairs

The creation of a synthetic base pair, expanding the canonical repertoire described by Watson and Crick [76], has been explored in various ways. Important for the in vivo incorporation of such base pairs into the DNA molecule is the transport into the cell, the enzymatic synthesis in the complex intracellular medium, and the tolerance towards the natural error elimination mechanisms. Retaining the information over replication cycles as well as retrieving the information in transcription and translation with high fidelity are fundamental requirements for a synthetic genetic information polymer [66]. This makes the modification of the components at the basis of the central dogma an extensive and challenging endeavor, which so far has not been developed successfully.

The four canonical base pairs rely on size and hydrogen-bonding complementation for their correct matching. Conserving these pairing rules may conserve the physicochemical character of the molecule [77], possibly allowing for an easier in vivo implementation. Chemical alteration of a canonical base pair is naturally only observed in bacteriophages, probably counteracting host restriction enzymes [78]. The research group of Benner pioneered the efforts to find additional base pairs relying on the canonical matching principles. While proving successful in vitro enzymatic polymerization [79] and replication [80] as well as stable transcription into RNA [81], the implementation into a living cell is so far difficult to achieve [82]. The directed evolution experiment of Marlière and colleagues aiming to exchange a canonical base in the entire genome of *E. coli* for a halogenated analog found that it is indeed possible to eliminate all of the natural thymidine down to the detection limit of 1.5% [83]. In their study, chlorouracil is carefully selected as precursor for intracellular conversion to a nucleotide, preserving the ultrastructure and hydrogen bonding characteristics of natural DNA. Although the resulting strain does not have expanded coding capabilities, DNA fragments containing the non-natural nucleotide will most likely be excluded from decoding by the canonical replication or transcription machinery of non-adapted cells.

Instead of relying on hydrogen bonding for base pairing, hydrophobic base pairs have been found useful for the in vivo expansion of the genetic alphabet. The Romesberg group successfully demonstrated the importance of exogenously phosphorylated unnatural nucleotides and their faithful in vivo replication in *E. coli* [84]. However, the plasmid-based replication of a single nucleotide at a specific position does not allow an evaluation for its usefulness in biocontainment. In later experiments, they demonstrated chromosomal incorporation and replication as well as assignment of the unnatural codon to an ncAA [85,86]. Thus far, this represents the most sophisticated platform of an in vivo genetic alphabet extension that has been developed. However, there have been legitimate concerns regarding the choice of the hydrophobic base pair which relies on stabilization of the surrounding canonical nucleotides for matching, possibly restricting the applicability for its extended use throughout the genome [87].

### 4.3. Alternative DNA Backbone Motifs

Synthetic DNA-like polymers (XNAs) have been engineered using non-natural sugar moieties or phosphate linkers of the DNA backbone, while conserving the four canonical bases. XNAs have been constructed with the ability to encode information, as well as bearing the potential for evolution [88]. XNAs are not readily processed by the natural replication and transcription enzymes [23], which makes it necessary to develop the according molecular machinery facilitating the intracellular processing of the genetic information. The same considerations as for the in vivo implementation of synthetic bases mentioned above are applicable for XNA molecules.

Threose nucleic acid (TNA) is a sugar-modified type of XNA that is able to bind to the reverse complement of RNA and DNA. Furthermore, it shows good stability in the intracellular milieu and a high resistance against natural nuclease enzymes [89]. The complementation of DNA and TNA single strands allows for the transcription of information from one molecule to the other, demonstrated through enzymatic transcription of DNA templates into TNA [90] and enzymatic reverse transcription of TNA templates into DNA [91]. In recent advances, the Chaput research group demonstrates a powerful approach computational approach for enzyme engineering, which was used to drastically improve TNA polymerase speed and accuracy. They used computational design together with sampling and pooling of beneficial mutations in a single enzyme, which leads to a an improved polymerase [92] able to polymerize TNA with an unnatural base [93]. However, the enzymes developed thus far lack orthogonality, and thus cannot easily be implemented in vivo, an issue concerning most of the developed XNAs [94]. 

Liu and colleagues recently published their progress towards the in vivo implementation of XNAs. They developed an XNA backbone modified in its sugar-phosphate moiety together with a cognate synthase with low affinity to natural nucleotides [95]. Although not demonstrating in vivo replication, their experiments show the poor acceptance of their XNA-DNA chimeras as a substrate for the endogenous *E. coli* replication machinery. However, it remains unclear if the proposed XNA chassis can be further developed to accommodate all necessary requirements to sustain life. The implementation of XNA molecules as genetic information storage in a cell-free system may certainly result in a robust biocontained system suitable for certain physically contained applications [28,77]. 

Complementing a minimal genome cell with an XNA based orthogonal central dogma, carrying contained information could be a fruitful strategy to achieve biocontainment in a precursor to true alien life [36].

## 5. The Farther the Safer

The premise “the farther the safer” [22], referring to the alienation of the core cellular chemistries which have naturally evolved under the universal genetic code, is still a legitimate claim if the aim is to eliminate the eventuality of genetic information transfer between naturally evolved organisms and GMOs.

The historical landscape of evolution with its specific dynamic environmental boundary conditions [96,97,98] and advantageous mechanisms of information transfer [20] has formed a narrow trajectory which seems to restrict the possibilities of cellular biochemistries to be explored [69]. Overcoming these restrictions by altering the meaning and identity of the genetic code will allow for drastically divergent evolutionary paths unable to exchange genetic information with, or revert back to, the naturally evolved counterpart (Figure 1).

The route to this new synthetic world will lead through vastly uncharted territory. There are two conceivable routes. Synthetic life can either be reached through a bottom up de novo cell design [30], or through the top down introduction of multiple ncAAs and the successive exchange of components of complex cellular biochemistry [31]. Promising progress has been reported towards de novo cell synthesis [99,100]. In the scope of this review we will however only discuss the top-down approach.

Although nucleotide and amino acid stereochemistry is decoupled in ribosomal protein biosynthesis through the use of tRNAs, it is the amino acid repertoire which shaped the chemical identity of the genetic code [34]. Organisms with an artificial amino acid repertoire thus far only maintain their installed biocontainment if essential proteins are dependent on the physicochemical character and related functional features of these synthetic building blocks. The stabilization of the topology changes necessary to accommodate the synthetic building blocks on a proteome scale remains the limiting factor, demonstrating the robustness of the naturally evolved cellular organization. 

Yu and colleagues [59] suspect that the safeguards against the proteome-wide substitution of a canonical amino acid with a close analog are encrypted in only a few cellular components and depend on the amino acid and the analog. Their study shows that the correct selective pressure does allow for the propagation of mutations strengthening an alternative genetic code. The organism which is auxotroph for tryptophan abolishes the uptake mechanism of the amino acid, a mutation that would otherwise be fatal. This somehow fragile containment of VGT could be strengthened by restricting the promiscuity of an engineered aaRS to only incorporate the ncAA, a strategy which has been proposed earlier by Bohlke and Budisa [61]. The topology changes surfacing in such a strain during a laboratory evolution experiment would certainly be interesting to observe. Until now, a number of organisms have been adapted to a variety of environmental conditions or equipped with desired metabolic traits through directed evolution approaches, all within moderate timeframes [101]. However, the insights necessary to establish such deep-rooted chemical changes like a sense codon reassignment are lacking, which impedes the design of appropriate starting or selection conditions for such evolutionary experiments [102]. More information on the topic can be retrieved from a recent review, which accumulates the extensive knowledge concerned with the knobs and dials of modern laboratory evolution [101].

Beyond facilitating HGT, the universal genetic code is thought to limit the adverse effects of information misinterpretation which may occur during the processes along the central dogma [103]. This feature usually complicates the stabilization of alternative genetic codes. However, there do exist some noteworthy exceptions of organisms or organelles having evolved deviating codon assignments [104,105,106]. For instance, *Micrococcus luteus* has been identified as an organism lacking six codon assignments, which could be exploited for the incorporation of ncAAs into proteins, or as a starting point for laboratory evolutions [22,107]. Similarly, the evolutionary path of mitochondria, which have lost a large amount of their cellular complexity while also maintaining an alternative codon assignment, could inform engineering approaches aiming to establish alternative genetic codes [108]. To date, the exact mechanism of how the assignment of a codon can change remains elusive. Recent studies predict the loss of a tRNA [54], or its deactivation due to a post-transcriptional modification, as the main drivers [109]. Detailed investigations into the mechanisms influencing these naturally occurring deviations from the standard genetic code might surface effective protocols with which to sustainably gain control over the process in laboratory evolutions.

## 6. Alienation Far beyond the Canonical Chemistries of Life

Life on earth is limited to the repertoire of 20 amino acids, which ensures protein function through reoccurring structural motifs based on hydrogen bonding. Thereby it becomes apparent that proline and glycine usually not participate in the β-sheet and α-helix secondary structures, but break these ordered formations to form loops [110]. This architecture has been extensively explored by nature through combinations of ordered formations as well as through post-translational modifications. Therefore, a long-term perspective would be to seek for other scaffolds that do allow a functional proteome to be based on different chemical architectures and alternative principles of protein folding. The creation of a ‘fundamentally new’ alien life would therefore require the use of radically new building blocks (and not the mere modifications of existing ones). Attributes and perspectives of alternative genetic codes whose repertoires are based on derivatives of proline [111], sarcosine, ornithine and other ‘alien’ building blocks are most recently elaborated by Kubyshkin and Budisa (Trends in Biotechnology, 2019, under revision).

If codon diversity and availability can be identified as a major hurdle to establish new chemistries into complex organisms, multiple-sense codon reassignments, quadruplet codons or XNAs with alternative base pairs might be the entry point for the sustainable incorporation of multiple synthetic building blocks into complex organisms. However, as explained above, the degeneracy of the genetic code mitigates potential adverse effects of genetic code misinterpretation. How technologies like quadruplet codons or sense codon reassignments affect the viability of cells in this regard remains to be determined. A synthetic organism based on an alien genetic code with a decoding logic incompatible to naturally evolved organisms, will be excluded from any HGT (receiving and transmitting). As XNA molecules have been shown to be capable of heredity and evolution [88], XNA-based organisms seem like a plausible vehicle for implementing proteome wide chemical changes. However, there is no experimental evidence that these organisms would absolutely depend on xeno-nutrients to survive, and thus VGT may not automatically be tightly controlled [22]. In the future, stronger, standardized tests have to be applied to newly created xeno-organisms with estranged genetic codes in order to quantify the robustness of the engineered containment against VGT, HGT and genetic drift in complex, diverse environments [13]. Only a synthetic organism that has been thoroughly tested for its absolute dependency on an otherwise inaccessible synthetic compound will be safe to employ in any complex open environment as long as no parts are toxic for any biological system in contact. However, depending on the time of interaction between the synthetic and biological world, an evolutionary drift towards preserving the genetic material and its contained information might be possible. Thus, for now, the attribute robust biocontainment cannot be assigned to any of the here presented technologies.

## Figures and Tables

**Figure 1 genes-10-00017-f001:**
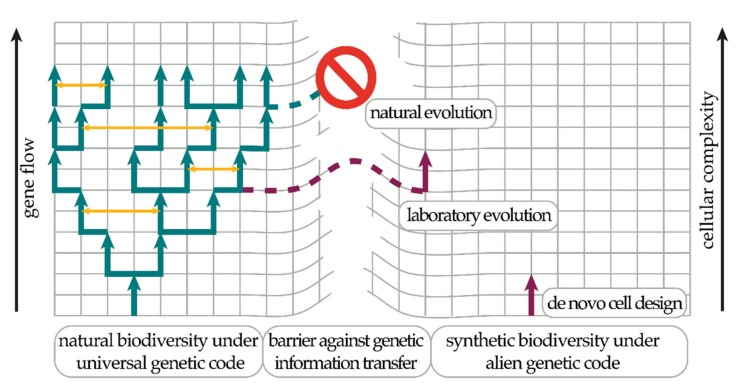
Biocontainment based on an alien genetic code. Life on Earth is a unity due to the existence of the universal genetic code. The exchange of genetic information from very different paths of evolution (horizontal gene transfer (HGT), yellow arrows) is facilitated by the universal genetic code fixing the basic core cellular chemistries (left side). Genetic information persists in time within a species through reproduction (vertical gene transfer (VGT), green arrows). A robust biocontainment system needs to restrict the flow of genetic information, which can be achieved through an alternative genetic code allowing for the exploration of drastically different cellular chemistries (right side). Currently, there are two promising experimental routes towards this goal: de novo (from scratch) cellular design (bottom-up) [30,31] or the successive alienation of a complex natural organism through laboratory evolution (top-down) [32,33].

**Figure 2 genes-10-00017-f002:**
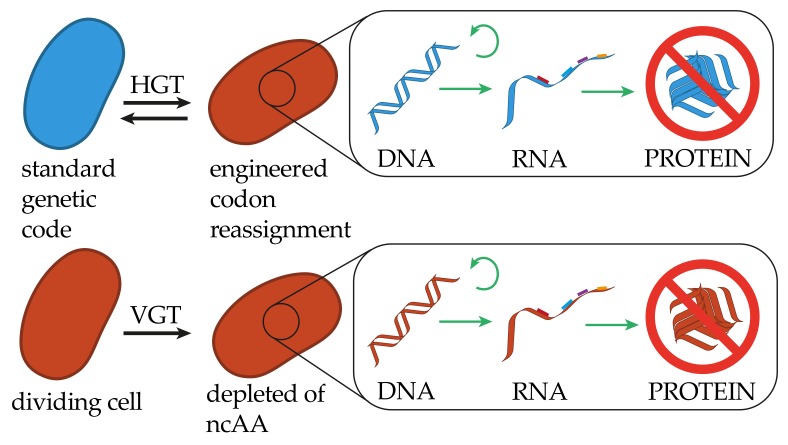
Restrictions of genetic information transfer in altering the meaning of the genetic code. In altering the codon assignment, cells receiving genetic information will be able to truthfully replicate, transcribe and translate the encoded information, however, the resulting polypeptides will not be functional. The same holds true for the genetic information transfer from genetically modified organisms (GMOs) to cells maintaining a standard genetic code. An engineered organism with a strict dependency for a non-canonical amino acid (ncAA) will not be capable of VGT in the absence of the ncAA.

**Figure 3 genes-10-00017-f003:**
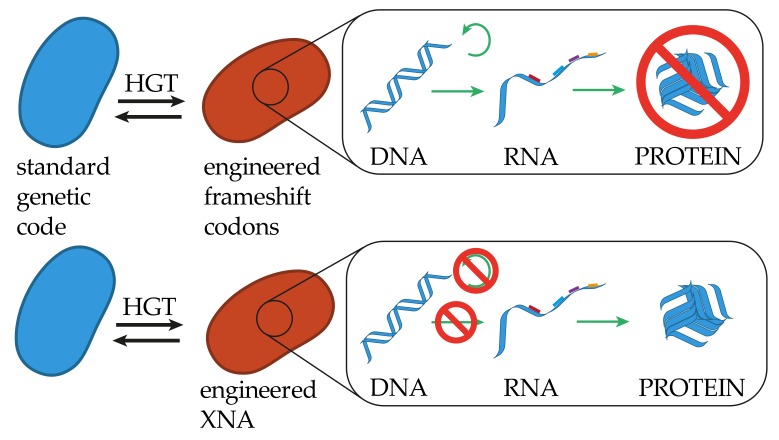
Restrictions of genetic information transfer in altering the identity of the genetic code. Similar to altering the canonical codon assignment, cells maintaining an expanded coding capacity, like quadruplet codons, will be able to replicate transcribe and translate genetic information received from a cell with a standard genetic code, where translation will not result in functional proteins. An organism strictly utilizing some kind of XNA backbone, while maintaining the canonical bases for genetic information storage, will not be able to replicate or transcribe genetic information in the form of DNA, but most likely be able to translate natural RNA into functional proteins.

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
