# Peer review of "Alternative Biochemistries for Alien Life: Basic Concepts and Requirements for the Design of a Robust Biocontainment System in Genetic Isolation"

_genes, 2018, doi:10.3390/genes10010017_

Round 1
Reviewer 1 Report
In this manuscript Diwo and Budisa provide a very interesting account of multiple strategies of engineering and evolving organisms towards the use of non-natural amino-acids, genetic codes, and DNA-like structures as a way to decrease the change of unwanted propagation of genetic information. I enjoyed reading this work and I it serves as a solid contribution to the field which many readers would gladly read. I have some small issues the authors might want to consider before publication.
1. Several of the phrasings in the manuscript are a bit clunky and difficult to read or understand. The authors might want to try to rephrase and clarify these. For example: “artificial alternation” (line 31), “physicochemical representation and decoding rules (logic)” (line 56), “Life-encoding events” (Line 88), “estranged genetic code” (Line 93), ‘molecular output’ (Line 100), “information transfer not successful” (Line 184), “exclusively encoding” (Line 187), “strive on several...” (Line 202), “at the command” (Line 274), and “encrypted for non-adapted cells” (Line 309).
2. The authors use several technical terms without proper explanation (‘word-dropping’), which could be quite confusing for the non-specialist reader. The authors might want to explain these in more details. For example: “system failure due to genetic drift” (Line 65), “1 escape in a population” (Line 67), “contained WT E. coli” (Line 163), and “mutation load” (Line 410)
3. The authors mentioned few times the bottom-up approach / de novo cell design but without clear explanation regarding what it exactly entail. This might deserve some more explanation.
4. I find it quite difficult to understand Figure 1 (the right side of it) and its take-home message.
5. A very promising way to achieve biocontainment is to replace living organism with cell-free systems composed of only enzymes and cofactors. The authors might want to relate to this emerging alternative.
6. The distinction between synthetic biology and xenobiology is not really convincing. In fact, xenobiology is hardly recognized as its own field and instead what the author describe as xenobiology is usually regarded as ‘classic’ synthetic biology. Does the distinction between synthetic biology and xenobiology really serve a purpose in this manuscript?
7. Isn’t “threat for the life on our planet, as catastrophic failure...” (Line 59) a bit to dramatic?
8. “Reducing the network complexity through decreasing the genetic code degeneracy” (Line 229-230): is “network” the right terminology?
9. “The standard genetic code confers an optimal solution to cellular organization” (Line 392): Really? On what basis? Isn’t it mainly a frozen accident?
10. “terrestrial organisms (i.e. life) are limited to an amino acid repertoire almost entirely developed from alanine derivatives, which ensure protein function through reoccurring structural motifs based on hydrogen bonding (Alanine World)” (Lines 413-415): this is beyond my understanding; are lysine, arginine, proline, phenylalanine, tyrosine, tryptophan, and histidine (for example) based on alanine? This does not make sense to me. The authors should explain this carefully and provide the appropriate references. Similarly, “proline, glycine and pyrrolysine can been seen as ‘strangers’” (Line 421): half the natural amino-acids are ‘strangers’ according to this definition...
Author Response
Dear Dr. O'Donoghue,
Thank you for your kind e-mail from December 13, 2018 including the comments of two reviewers. We revised our manuscript according to all the questions, suggestions and comments of all reviewers. At this point, we would also like to kindly thank to referees for their impartial judgement and good suggestions, which has contributed to a better presentation of our work in the revised manuscript.
With these changes, we hope to meet all the referees’ requirements and to fulfil all standards of your renowned journal.
We are looking forward to hearing from you!
Sincerely yours,
Christian Diwo and Nediljko Budisa
On the following pages, we list point-by-point the detailed changes made to the manuscript according to all reviewer´s suggestions. All changes have been included with the word tracking feature. We are confident that our revisions will help avoid possible misunderstandings and improve the quality of the manuscript.
Referee 1:
Several of the phrasings in the manuscript are a bit clunky and difficult to read or understand. The authors might want to try to rephrase and clarify these. For example: “artificial alternation” (line 31), “physicochemical representation and decoding rules (logic)” (line 56), “Life-encoding events” (Line 88), “estranged genetic code” (Line 93), ‘molecular output’ (Line 100), “information transfer not successful” (Line 184), “exclusively encoding” (Line 187), “strive on several...” (Line 202), “at the command” (Line 274), and “encrypted for non-adapted cells” (Line 309).
The authors use several technical terms without proper explanation (‘word-dropping’), which could be quite confusing for the non-specialist reader. The authors might want to explain these in more details. For example: “system failure due to genetic drift” (Line 65), “1 escape in a population” (Line 67), “contained WT E. coli” (Line 163), and “mutation load” (Line 410)
We thank the referee for these suggestions. We agree that in the wording in the original manuscript was in parts challenging which may have led to misunderstanding. In the revised manuscript we followed the advice of this referee and rephrased some passages. In addition, all mentioned terms have been more thoroughly described. We believe that this will contribute significantly to increase the comprehensibility of our manuscript. All changes are marked in the text of the revised Ms.
3. The authors mentioned few times the bottom-up approach / de novo cell design but without clear explanation regarding what it exactly entail. This might deserve some more explanation.
We thank the referee for these suggestions. However, we believe that detailed description of bottom-up approach / de novo cell design would go beyond the scope of this manuscript. It should be stressed that we are mainly concerned with alienation of the genetic code through laboratory evolution. On the other hand, synthetic biology of bottom-up/ top down approaches are excellently covered in the contemporary literature. Thus, we provided two most prominent literature references in the caption of Figure 1 as well as in the text of the revised manuscript (line 545).
4. I find it quite difficult to understand Figure 1 (the right side of it) and its take-home message.
We appreciate the comment and agree that a better explanation should be provided. Therefore, we fully rephrased the figure caption in the revised Ms by separately describing the left and right part of the graph.
5. A very promising way to achieve biocontainment is to replace living organism with cell-free systems composed of only enzymes and cofactors. The authors might want to relate to this emerging alternative.
We appreciate this relevant comment and thank the referee for bringing this issue to our attention. Firstly, the focus of our study is on the alienation of life (as autonomously replicating biochemical systems) and the deployment of such alienated life forms in open environments. Secondly, switching to cell-free systems could be considered as biosafety approach, although the presence of genetic material also requires precautionary measures, as DNA fragments of organisms can also be taken up and HGTs could be produced as well. Finally, cell-free systems with XNAs as genetic polymers might be the ultimate biosafety tool much safer compared to those with natural polymers. We elaborated shortly on this issue in our revised Ms (lines 524-526) to emphasize these opportunities.
6. The distinction between synthetic biology and xenobiology is not really convincing. In fact, xenobiology is hardly recognized as its own field and instead what the author describe as xenobiology is usually regarded as ‘classic’ synthetic biology. Does the distinction between synthetic biology and xenobiology really serve a purpose in this manuscript?
We thank the referee for bringing this issue to our attention. In our opinion, the "classical" synthetic biology assumes the modularity of life, i.e. every life form in a certain combination of modules. Living beings of the past and present are selected ("filtered") from specific environments because their specific modular organization was best suited for survival. The "classical" synthetic biology also assumes that nature has not exploited all possible combinations of these modules. This is the logic behind the claim of Drew Endy that one could reprogram an oak tree seed to grow into a living wooden house. Finally, the "classical" synthetic biology takes the basic chemical make-up of living cells as invariant (life on earth is indeed chemically invariant), while xenobiology will change that. The term Chemical Synthetic Biology is also present in the literature albeit not widely used.
Although distinction between synthetic biology and xenobiology is not a purpose in this manuscript, we believe that it is necessary to emphasize this stage of the field development. Clearly, it is still difficult to distinguish between these two fields but future developments will certainly leverage XB as stand-alone research field.
In the revised manuscript (lines 41-73) we elaborated on these issues.
7. Isn’t “threat for the life on our planet, as catastrophic failure...” (Line 59) a bit to dramatic?
Changed in the revised manuscript (line 86-88).
8. “Reducing the network complexity through decreasing the genetic code degeneracy” (Line 229-230): is “network” the right terminology?
We appreciate this comment and provided better explanation in our revised manuscript
(lines 371-373).
9. “The standard genetic code confers an optimal solution to cellular organization” (Line 392): Really? On what basis? Isn’t it mainly a frozen accident?
We thank the referee for this comment. In the revised manuscript we argued that besides being a “frozen accident” the genetic code present a solutions to translational error minimizations which was conceptualized already in 1960s immediately after the code deciphering (e.g. Sonneborn, 1966). However, the definition as optimal solution does not hold true in this context without a more elaborate discussion, so we changed the wording (line 629-631).
10. “terrestrial organisms (i.e. life) are limited to an amino acid repertoire almost entirely developed from alanine derivatives, which ensure protein function through reoccurring structural motifs based on hydrogen bonding (Alanine World)” (Lines 413-415): this is beyond my understanding; are lysine, arginine, proline, phenylalanine, tyrosine, tryptophan, and histidine (for example) based on alanine? This does not make sense to me. The authors should explain this carefully and provide the appropriate references. Similarly, “proline, glycine and pyrrolysine can been seen as ‘strangers’” (Line 421): half the natural amino-acids are ‘strangers’ according to this definition...
Starting from the fact that the canonical amino acids are forming secondary structures based on hydrogen bonding Kubyshkin et al found experimentally evidence that indeed secondary structures based on interactions other than hydrogen boding are possible. Therefore, it is plausible to hypothesize that life based on different chemical architectures and alternative underlying principles of protein folding is possible.
However, we agree that our statements in the original manuscript (due to the missing of solid experimental evidence) might be too speculative and misleading. In the revised Ms we avoid the use of the term “alanine world” and focus on the fact that contemporary attempts to change amino acid repertoire are not radical enough (“mere modifications of existing ones”) i.e. the need to go far beyond the existing underlying concepts.
Reviewer 2 Report
The review article by Christian Diwo and Nediljko Budisa “Alternative Biochemistries for Alien Life: Basic concepts and requirements for the design of a robust biocontainment system in genetic isolation” describes strategies for altering the genetic code for the purpose of creating robust biocontainment system. As our ability to create synthetic designer organism is increasing it is important that we pay more attention to the development of biocontainment systems.
Before the review article can be published some revisions will be required.
1. Abstract, lines 19- 21: “In order to gain control over all possible mechanisms of genetic information exchange, synthetic life has to be based on an alien genetic code conferring cellular chemistries that are incompatible to nature.” Authors should change “all possible mechanisms” to “most possible mechanisms” or similar statement as it would be extremely challenging to gain control over all possible mechanisms.
Similar changes will need to be made throughout the whole review if “all” is used.
2. Line 58-60. In the following text “It is however the universality of the genetic code which makes genetically modified organisms (GMOs) [20] a threat for the life on our planet, as catastrophic failure of the intended use might contaminate the naturally evolved genetic pool and alter entire ecosystems” the reference used (construction of biologically functional bacterial plasmids) does not support this overly dramatic statement. Please reword it and use appropriate citations to support your statements.
Authors should carefully check that all publications are properly cited through the whole review.
3) Lines 113-114: “The main vectors of HGT between prokaryotes are bacteriophages, plasmids or conjugation [32].” This statement should be reworded since HGT between prokaryotes can happen via transduction, conjugation or natural DNA uptake. In all cases, plasmids/or parts of plasmids can be transferred to the prokaryotes.
Authors need to pay attention to the correctness of their statements.
4) Authors talk about Minimal cell Design (line 225) but they have not mentioned how the recently built minimal cell could be used for this approach ( Hutchison, C. A., et al (2016) Design and synthesis of a minimal bacterial genome. Science, http://science.sciencemag.org/content/351/6280/aad6253)
5) Robust biocontainment systems are an absolute necessity as scientists create more GMOs, therefore, it would be good if the authors could comment if any of the proposed designs could actually work in places such as human gut microbiome, soil rhizosphere, oceans etc. Would organisms with such alien genetic code be robust enough to survive in nature? Or is the goal to create GMOs that will be always contained in fermentation tanks or similar environments.
Author Response
Manuscript ID: genes-405335
"Alternative Biochemistries for Alien Life: Basic concepts and requirements for the design of a robust biocontainment system in genetic isolation"
Response to Reviewers:
Dear Dr. O'Donoghue,
Thank you for your kind e-mail from December 13, 2018 including the comments of two reviewers. We revised our manuscript according to all the questions, suggestions and comments of all reviewers. At this point, we would also like to kindly thank to referees for their impartial judgement and good suggestions, which has contributed to a better presentation of our work in the revised manuscript.
With these changes, we hope to meet all the referees’ requirements and to fulfil all standards of your renowned journal.
We are looking forward to hearing from you!
Sincerely yours,
Christian Diwo and Nediljko Budisa
On the following pages, we list point-by-point the detailed changes made to the manuscript according to all reviewer´s suggestions. All changes have been included with the word tracking feature. We are confident that our revisions will help avoid possible misunderstandings and improve the quality of the manuscript.
Referee2:
The review article by Christian Diwo and Nediljko Budisa “Alternative Biochemistries for Alien Life: Basic concepts and requirements for the design of a robust biocontainment system in genetic isolation” describes strategies for altering the genetic code for the purpose of creating robust biocontainment system. As our ability to create synthetic designer organism is increasing it is important that we pay more attention to the development of biocontainment systems.
Before the review article can be published some revisions will be required.
1. Abstract, lines 19- 21: “In order to gain control over all possible mechanisms of genetic information exchange, synthetic life has to be based on an alien genetic code conferring cellular chemistries that are incompatible to nature.” Authors should change “all possible mechanisms” to “most possible mechanisms” or similar statement as it would be extremely challenging to gain control over all possible mechanisms.
We appreciate the concern raised by the referee that points out an important issue. We have changed the text of the revised Ms accordingly. Our intention is indeed to gain control of all known mechanisms of genetic information exchange (VGT, HGT). The circumvention of these mechanisms by some unknown process cannot be excluded. We are arguing (in the last part of the revised manuscript) that evolutionary time needed to extract information from XNA molecules would be too short in the frame of standard experimental scenario/set-up.
Similar changes will need to be made throughout the whole review if “all” is used.
We changed the context accordingly.
2. Line 58-60. In the following text “It is however the universality of the genetic code which makes genetically modified organisms (GMOs) [20] a threat for the life on our planet, as catastrophic failure of the intended use might contaminate the naturally evolved genetic pool and alter entire ecosystems” the reference used (construction of biologically functional bacterial plasmids) does not support this overly dramatic statement. Please reword it and use appropriate citations to support your statements.
We appreciate the comment and agree that a better explanation should be provided. The reference was related to the term GMO (first time introduced in the manuscript). The statement has been rephrased with appropriate literature references cited (line 85-87).
Authors should carefully check that all publications are properly cited through the whole review.
Done.
3) Lines 113-114: “The main vectors of HGT between prokaryotes are bacteriophages, plasmids or conjugation [32].” This statement should be reworded since HGT between prokaryotes can happen via transduction, conjugation or natural DNA uptake. In all cases, plasmids/or parts of plasmids can be transferred to the prokaryotes.
We appreciate this relevant comment and thank the referee for bringing this to our attention. We reworded the original statement in the revised manuscript (lines 203).
Authors need to pay attention to the correctness of their statements.
Done.
4) Authors talk about Minimal cell Design (line 225) but they have not mentioned how the recently built minimal cell could be used for this approach ( Hutchison, C. A., et al (2016) Design and synthesis of a minimal bacterial genome. Science, http://science.sciencemag.org/content/351/6280/aad6253)
We thank the referee for this comment. This information is added in the paragraph (lines 365-379) discussing minimal genome, in addition to discussion of reducing the degeneracy of the genetic code for minimal cells design and their use in biocontainment.
5) Robust biocontainment systems are an absolute necessity as scientists create more GMOs, therefore, it would be good if the authors could comment if any of the proposed designs could actually work in places such as human gut microbiome, soil rhizosphere, oceans etc. Would organisms with such alien genetic code be robust enough to survive in nature? Or is the goal to create GMOs that will be always contained in fermentation tanks or similar environments.
We appreciate this relevant comment and have expanded our discussion in the last chapter of the revised manuscript (lines 82-831).